# Chemical Synthesis of a Functional Fluorescent-Tagged α-Bungarotoxin

**DOI:** 10.3390/toxins14020079

**Published:** 2022-01-21

**Authors:** Oliver Brun, Claude Zoukimian, Barbara Oliveira-Mendes, Jérôme Montnach, Benjamin Lauzier, Michel Ronjat, Rémy Béroud, Frédéric Lesage, Didier Boturyn, Michel De Waard

**Affiliations:** 1L’institut Du Thorax, Nantes Université, INSERM, CNRS, INSERM UMR 1087/CNRS UMR 6291, 8 Quai Moncousu, F-44007 Nantes, France; oliver.brun@univ-nantes.fr (O.B.); Barbara.Ribeiro@univ-nantes.fr (B.O.-M.); jerome.montnach@univ-nantes.fr (J.M.); benjamin.lauzier@univ-nantes.fr (B.L.); michel.ronjat@univ-nantes.fr (M.R.); 2Faculty of Medicine, Université de Montréal, Montréal, QC H3T 1J4, Canada; 3Research Center, Montréal Heart Institute, Montréal, QC H1T 1C8, Canada; frederic.lesage@polymtl.ca; 4Smartox Biotechnology, 6 Rue des Platanes, F-38120 Saint-Egrève, France; claude.zoukimian@smartox-biotech.com (C.Z.); remy.beroud@smartox-biotech.com (R.B.); 5École Polytechnique Montréal, Montreal, QC H3C 3A7, Canada; 6UMR 5250, Département de Chimie Moléculaire, CNRS, Université Grenoble Alpes, CEDEX 09, F-38058 Grenoble, France; didier.boturyn@univ-grenoble-alpes.fr; 7LabEx “Ion Channels, Science & Therapeutics”, F-06560 Valbonne, France

**Keywords:** toxins, peptide chemistry, native chemical ligation, α-bungarotoxin, click chemistry, automated patch-clamp, fluorescent peptide, TE671 cells, nicotinic acetylcholine receptor

## Abstract

α-bungarotoxin is a large, 74 amino acid toxin containing five disulphide bridges, initially identified in the venom of *Bungarus multicinctus* snake. Like most large toxins, chemical synthesis of α-bungarotoxin is challenging, explaining why all previous reports use purified or recombinant α-bungarotoxin. However, only chemical synthesis allows easy insertion of non-natural amino acids or new chemical functionalities. Herein, we describe a procedure for the chemical synthesis of a fluorescent-tagged α-bungarotoxin. The full-length peptide was designed to include an alkyne function at the amino-terminus through the addition of a pentynoic acid linker. Chemical synthesis of α-bungarotoxin requires hydrazide-based coupling of three peptide fragments in successive steps. After completion of the oxidative folding, an azide-modified Cy5 fluorophore was coupled by click chemistry onto the toxin. Next, we determined the efficacy of the fluorescent-tagged α-bungarotoxin to block acetylcholine (ACh)-mediated currents in response to muscle nicotinic receptor activation in TE671 cells. Using automated patch-clamp recordings, we demonstrate that fluorescent synthetic α-bungarotoxin has the expected nanomolar affinity for the nicotinic receptor. The blocking effect of fluorescent α-bungarotoxin could be displaced by incubation with a 20-mer peptide mimicking the α-bungarotoxin binding site. In addition, TE671 cells could be labelled with fluorescent toxin, as witnessed by confocal microscopy, and this labelling was partially displaced by the 20-mer competitive peptide. We thus demonstrate that synthetic fluorescent-tagged α-bungarotoxin preserves excellent properties for binding onto muscle nicotinic receptors.

## 1. Introduction

α-bungarotoxin is a 74-mer peptide containing five disulphide bridges. It was originally identified and purified from the venom of the snake *Bungarus multicinctus*, an elapid snake from Taiwan [1]. Its unique properties were soon recognized through its ability to act as an antagonist of acetylcholine (ACh) for the nicotinic acetylcholine receptor (AChR) on which it binds, mainly by interacting with the α subunits of AChR with very high affinity and irreversibly [2]. As member of the type II α-neurotoxins (five disulphide bridges as opposed to type I with four disulphide bridges), it belongs to the family of three-finger toxins. It binds both the muscle type AChR (mAChR) at the motor endplate and the neuronal α7 AChR. While two and five binding sites are available, respectively, for these two types of AChR, respectively, the binding of a single α-bungarotoxin is sufficient to block channel opening of the ligand-gated channels, indicating a non-competitive mechanism of antagonism [3]. Structural data reveal that glycosylation of AChR is important for the high-affinity binding of the toxin [4]. Among all the interesting applications developed with this toxin are the improved long-term imaging of zebrafish embryo development using genetically encoded α-bungarotoxin, allowing for a complete immobilization of the embryo [5]. While label-free α-bungarotoxin-binding assays have been developed, namely using a BIAcore sensor chip technology [6], most applications require labelling of the toxin. Very early on, the purified peptide could be iodinated to characterize the properties of the toxin receptor at the level of the skeletal muscle [7]. This allowed for a precise quantification of mAChR numbers at the motor end plate of rat diaphragm [8]. As usual for toxins employed as radiotracers, the iodinated version of α-bungarotoxin allowed tracking the purification of AChR from *Torpedo californica* electroplax [9]. The scope of investigation on AChR being very dynamic, it did not take long before an FITC-, TRITC- or rhodamine-derived fluorescent version of the purified α-bungarotoxin was identified [10] and used to visualize the distribution of AChR onto vertebrate skeletal muscle fibres [11,12]. Most of this labelling involves N-hydroxysuccinimide (NHS) ester-mediated derivatisation of an amine-reactive group of α-bungarotoxin. Interestingly, in vivo injection of fluorescent α-bungarotoxin was shown to improve the efficiency of motor endplate labelling [13]. Yet, it was clearly recognized that, depending on the ratio of dye/toxin, this labelling decreased the blocking potency of α-bungarotoxin for AChR [11], since fluorescence labelling occurs randomly along the sequence onto free amines, some possibly required for pharmacophore integrity. Similarly, a method was presented for the efficient conjugation of horseradish peroxidase to α-bungarotoxin, which is useful for the histochemical staining of mAChR on muscle fibres for light and electron microscopy [14] and for the coupling to gold particles for electron microscopy applications [15]. Yet, in spite of a claimed 1:1 molar conjugation, labelled α-bungarotoxin seems to lose part of its activity. The importance of tagged α-bungarotoxin is nevertheless substantiated by the fact that the α-bungarotoxin binding site (BS), an optimized small peptide of about 13 amino acids derived from a combinatorial phage-display peptide library, fully preserves the unique properties of high affinity binding of the toxin [16,17]. Hence, in addition to the fact that it may serve as a lead compound for the development of antidotes [18], this BBS peptide can also be used as a tag onto poorly immunogenic proteins to visualize their cell behaviour with tagged α-bungarotoxin [19,20,21,22,23,24,25,26,27,28]. 

Most of the applications reporting α-bungarotoxin usage have been performed with the toxin purified directly from snake venom. There are two reasons for this: (i) first, high quantities of this venom can be obtained by milking this snake species, and (ii) α-bungarotoxin is abundantly represented in this venom. As a consequence, it took quite some time before anyone published a method for the production of recombinant α-bungarotoxin [28]. However, recombinant α-bungarotoxin comes with additional amino acids at the N-terminus. Efforts to produce synthetic α-bungarotoxin have also been delayed considerably because of the complexity of the chemical synthesis; the size of the peptide is substantial, as is the number of disulphide bridges. Yet, this effort is crucial if one desires to produce analogues containing non-natural amino acids or if site-defined tagging onto the toxin is the goal. A recent study reported the first chemical synthesis of α-bungarotoxin using peptide-hydrazide-based native chemical ligation (NCL) between two α-bungarotoxin fragments [29]. While this is an important achievement, there was no conclusive evidence in this report that the synthetic α-bungarotoxin was functionally active on mAChR or that this chemical synthesis could be used to produce fluorescent analogues. Herein, our aim was to produce synthetic α-bungarotoxin with an alkyne function at the N-terminus of the chain in order to selectively conjugate any fluorophore or molecule of interest like biotin via click chemistry. With regard to the large size of this toxin, we decided to synthesize the linear chain thanks to two successive ligations of three α-bungarotoxin fragments. In addition, we kept the hydrazide-based ligation strategy to synthesize the linear α-bungarotoxin, as it can be considered the most convenient native chemical ligation method and currently is one of the most used [30]. The oxidative folding leads to folded α-bungarotoxin functionalized with an alkyne group. We performed a wealth of functional tests to illustrate that tagged synthetic α-bungarotoxin maintains potent activity compared to the native purified peptide. Our data illustrate that we capably produced for the first time a synthetic fluorescent α-bungarotoxin that preserves all the functionalities of purified α-bungarotoxin, opening an avenue for new applications that require specific sequence peptide toxin modifications, which are not possible on the natural peptide or only partially on the recombinant peptide. 

## 2. Results and Discussion

### 2.1. Strategy of Synthesis and Chemical Production of Intermediate Fragments of α-Bungarotoxin

The amino acid length of α-bungarotoxin is too long to allow a full one-shot peptide synthesis by solid-phase peptide synthesis (SPPS). For this reason, we first selected three α-bungarotoxin peptide fragments with sizes most favourable for performing peptide-hydrazide-based native chemical ligation [31] by considering the different ligation sites (Xaa-Cys, Xaa being any kind of amino acid) available (Figure 1). Three fragments turned out to be favourable for this synthesis approach. PA-α-BgTx_1-28_ (**1**; IVCHTTATSPISAVTCPPGENLCYRKMW) with a pentynoic acid (PA) at the amino terminus appears suitable because it offers a Trp-Cys point of ligation. The alkyne moiety instead of its azide counterpart was chosen for later click chemistry in order to avoid the reduction of the azide function that may potentially occur during oxidative folding of full-length α-BgTx. A shorter α-BgTx_1-22_ fragment was also possible to synthesize (**2**; IVCHTTATSPISAVTCPPGENL) but the ligation point Leu-Cys is less favourable [32]. The intermediate fragment α-BgTx_29-43_ (**3**; CDAFCSSRGKVVELG) was preferred over the lengthier fragment α-BgTx_29-47_ (**4**; CDAFCSSRGKVVELGCAAT) because it provides an ideal Gly-Cys ligation point despite a shorter length than **4**. The third fragment was α-BgTx_44-74_ (**5**; CAATCPSKKPYEEVTCCSTDKCNPHPKQRPG). The first two fragments were synthesized with a hydrazide function at the C-terminus in order to achieve two successive ligations. The use of hydrazide fragments, known as crypto-thioesters (i.e., masked peptide thioesters, which, unless activated, are inert in NCL), allows the ligation to occur in the N-to-C direction without requiring temporary orthogonal protection of the Cys residues, as in the case of the Dawson method [33]. While o-aminoanilide fragments [34] could also have been used as masked thioesters, this would have implied coupling a Gly residue as the first amino acid onto a Dawson resin with the α-BgTx_29-43_ fragment. This reaction can easily acylate the two amine functions of this resin, even if a second generation of the Dawson resin has addressed this issue [35].

The syntheses of these three fragments (**1**, **3** and **5**) were performed by automated SPPS, yielding crude material with clear major analytical reversed-phase high-performance liquid chromatography (RP-HPLC) peaks and yields after purification of 26% (**1**), 57% (**3**) and 44% (**5**) (Figure 2A–C). The products had the appropriate masses (insets in Figure 2).

### 2.2. Chemical Synthesis of Full-Length α-Bungarotoxin

Our initial aim was to assemble the linear α-BgTx peptide starting from the N- to the C-terminus by ligating PA-α-BgTx_1-28_ (**1**) with α-BgTx_29-43_ (**3**) to obtain PA-α-BgTx_1-43_ (**6**), followed by PA-α-BgTx_1-43_ ligation with α-BgTx_44-74_ (Figure 1A). This was attempted by first selectively activating the hydrazide function of **1** with NaNO_2_ (15 min, −20 °C), which was later neutralized by an excess of MPAA. The subsequent addition of fragment **3** leads to the expected intermediate fragment **6**, PA-α-BgTx_1-43_, by NCL after one night (Figure 1A). Unfortunately, the first ligation leads to a mixture of the desired product, with truncated peptides corresponding to the hydrolysis of one or two amino acids before and after the ligation site, as determined by mass spectroscopy. High-resolution mass spectrometry (HRMS) analyses allowed the identification of the amide bonds that were cleaved within the peptide; they were all located at one or two residue distance from the ligation site, either towards the N-terminus (amide bond Met^27^-Trp^28^) or towards the C-terminus (amide bonds Asp^30^-Ala^31^ and Ala^31^-Phe^32^). Fragments **1** and **3** were not sensitive to the NCL buffer and, interestingly, when the NCL reaction was performed between fragments **1** and **5** (instead of **3**), the resulting peptide PA-α-BgTx_1-28,44-74_ was stable (data not shown). These controls indicate that the degradation of **6** is an intrinsic property of its sequence.

Based on the observation that such a degradation is not observed with the full length α-BgTx, we then decided to shift towards a second strategy by assembling the fragments from the C-terminus to the N-terminus (Figure 1B). This avoids the formation of the intermediate peptide **6**. Therefore, α-BgTx_29-43_ was synthesized again but with an Acm protection on the N-terminal cysteine of the fragment to yield Acm-α-BgTx_29-43_ (**7**) in order to selectively link this fragment to **5**, α-BgTx_44-74_ (Figure 1B). This protection is fully compatible with hydrazide-based NCL conditions [36] and prevents the cyclisation or polymerisation of the fragment in the NCL conditions once the hydrazide function is converted into a thioester. Following activation of the hydrazide function by NaNO_2_ (15 min, −20 °C), the NCL was conducted with fragments **7** and **5** at RT overnight and in the presence of MPAA. After purification, the ligation product **8** (Acm-α-BgTx_29-74_) was obtained with a yield of 60% and with the expected mass (Figure 2A). No degradation product was observed. Next, the Acm protecting group was removed from the N-terminal Cys residue of **8**. For that purpose, we used a published method based on the use of palladium and DTT [37]. Fragment **8** was solubilized in NCL buffer at neutral pH and incubated with 10 equivalents of PdCl_2_ for 10 min at 37 °C. Next, 100 equivalents of DTT were added to neutralize the excess palladium and eliminated by centrifugation before purification by RP-HPLC. Fragment **9** (α-BgTx_29-74_) was hence obtained pure, with a yield of 53% (Figure 2B). This last fragment could therefore be used for a second NCL with the PA-α-BgTx_1-28_ fragment (**1**); **1** was therefore first activated by oxidation with NaNO_2_ and later put in presence of MPAA and **9**. Following purification and lyophilisation, unfolded and reduced PA-α-BgTx (**10**) was obtained with a yield of 42% (Figure 2C). In order to oxidize all the ten cysteine residues into the five natural disulphide bonds, several oxidizing buffers were used. The alkaline buffer containing 0.1 M tris(hydroxymethyl)aminomethane (Tris), pH 8.2 and the redox couple reduced glutathione (GSH, 5 mM)/oxidized glutathione (GSSG, 0.5 mM) yielded the best results. After 48 h, and following RP-HPLC purification, the folded/oxidized PA-α-BgTx (**11**) was obtained with a yield of 23% (Figure 1B and Figure 2D). The total yield after 4 steps was 3%.

### 2.3. Coupling of a Fluorochrome on PA-α-BgTx by Click Chemistry

To check the feasibility of using click chemistry on **11**, we decided to selectively couple Cy5-N3 (**12**) onto **11** by copper-catalysed azide-alkyne cycloaddition (CuAAC) [38,39]. Following solubilization of **11**, 10 equivalents of **12**, 10 equivalents of CuBr(CH_3_)_2_S and 5 equivalents of tris(3-hydroxypropyltriazolylmethyl)amine (THPTA) were added and incubated at RT overnight and under inert atmosphere. Cy5-α-BgTx (**13**) was purified by RP-HPLC, with a yield of 21% based on the dosing at 646 nm of Cy5 emission (Figure 2E). The excess of dye **12** was also collected to be used again in further click reactions. To test whether **13** was bioactive, we launched a series of functional tests to validate the synthetic product.

### 2.4. High-Throughput Evaluation of ACh and Cy5-α-Bungarotoxin Effects on the TE671 Cell Line

Following chemical synthesis of Cy5-α-BgTx, the next step was to assess whether this compound retained the inhibitory effect and affinity for mAChR. This was performed by recording mAChR currents in TE671 cells elicited by ACh application in the presence of either native purified α-BgTx or synthetic Cy5-α-BgTx using a high-throughput automated patch-clamp system (Syncropatch 384, Nanion, Munich, Germany). This system allows the simultaneous recording of 384 cells, permitting a quantitative evaluation of the effects of ACh and peptides. The resting voltage was set to −60 mV to record inward currents through mAChR. The use of TE671 human cell line, derived from rhabdomyosarcoma, for this study is appropriate since it endogenously expresses mAChR [40] and binds α-BgTx [41]. Each cell was maintained at a resting membrane potential of −60 mV to record inward currents through mAChR. The effects of a first pulse of 3.33 µM ACh, in the absence of toxins, and of a second 3.33 µM ACh pulse following a 15 min incubation with one of the α-BgTx analogues were monitored. Only cells considered responsive, with a minimal current amplitude of −100 pA after the first ACh application, were selected for analyses in order to clearly discriminate the effects of the peptides. As expected from a channel that tends to desensitize the second application of ACh, it provided a lower current amplitude (Figure 3A). Indeed, the maximal current amplitude recorded at the second dose averaged −230.5 ± 22.9 pA versus −424.2 ± 45.0 pA for the first ACh pulse (*n* = 82; *p* < 0.005). Since this level of desensitization is strictly correlated with ACh concentration and time of application [42,43,44], we decided to normalise the current elicited by the second dose of ACh against that obtained for the first pulse using individual cells as their own intrinsic control, thereby accounting for intercellular variability. ACh concentration and time of application could not be further optimised, as they correspond roughly to the EC_50_ ligand’s value on mAChR [45] and a short 5 s respectively. Therefore, to assess the effects of native α-BgTx and synthetic Cy5-α-BgTx, the pulse 2/pulse 1 (P2/P1) ratio for each cell was normalised against the mean P2/P1 obtained for control cells in the absence of toxins (Figure 3B). Clear evidence was thus provided that both peptides, at concentrations above 0.03 nM, were able to block the response of TE671 cells to a second application of ACh (dashed arrows illustrating the reduced current levels from dotted horizonal lines). From the current amplitudes recorded using this protocol, concentration-response curves for purified α-BgTx and Cy5-α-BgTx were generated (Figure 3C). Cy5-α-BgTx potently blocks mAChR currents with an IC_50_ value of 1.64 ± 0.19 nM (*n* = 19–28 cells/concentration). In comparison, the natural purified and non-modified α-BgTx peptide inhibits mAChR with a slightly better potency, with an IC_50_ value of 0.25 ± 0.02 nM (*n* = 8–41 cells/concentration). The Hill coefficient remained unaltered by the addition of the Cy5 tag, with an estimation of 1.45 ± 0.17 for native α-BgTx and 1.38 ± 0.20 for Cy5-α-BgTx. Both affinity and Hill coefficient estimations are consistent with previous records of α-BgTx binding on TE671 cells [46]. The slight difference in IC_50_ values observed between native and synthetic modified α-BgTx can be accounted for by the difficulties in reliably assessing the concentrations of purified α-BgTx and the potential presence of the fluorescent tag at the N-terminus of the peptide. This is thus the first report illustrating the activity on its receptor of a synthetic α-BgTx modified at its N-terminus with a fluorescent tag.

### 2.5. Neutralisation of α-BgTx-Mediated Inhibition of ACh Response by a Binding Site-Mimicking Peptide

As discussed previously, peptides representing the mAChR binding site for α-BgTx are known, and a mimicking peptide can be synthesized. We added, to the optimum binding sequence (WRYYESSLEPYPD-OH), 3 amino acids (GSG) at both the N- and C-terminus, and an additional biotin tag onto an extra C-terminal K residue. The resulting peptide used for our experiments is GSGWRYYESSLEPYPDGSGK(biotin)-OH (termed BSpep). Due to the pre-established irreversibility of α-BgTx binding to mAChR, it was hypothesised than pre-incubating the toxin with the binding site analogue should neutralise it, thereby preventing its inhibition of ACh activity on the muscle-type mAChR. We first assessed the binding of variable concentrations of BSpep to 5 nM Cy5-α-BgTx by microscale thermophoresis (MST) (Figure 4). Fit of the data yields an apparent K_d_ of 1.04 µM.

Next, the neutralising effect of BSpep was assessed in vitro on TE671 cells using the aforementioned patch-clamp assay. 1 μM BSpep was preincubated with 3.3 nM native α-BgTx or 10 nM Cy5-α-BgTx at 4 °C for 2 h, prior to incubation with the TE671 cells for 15 min. Incubation with 1 µM BSpep completely restored normal ACh response in the presence of maximally blocking concentrations of either natural purified α-BgTx or synthetic Cy5-α-BgTx (Figure 5A). Inhibition with 3.33 nM α-BgTx resulted in a complete blockade of the ACh-mediated current, reaching only 0.1 ± 2% of the control current elicited by ACh alone (*n* = 13) (Figure 5B). However, preincubation of the natural α-BgTx peptide with 1 μM BSpep rescued currents up to 100 ± 6% of the normal current (*n* = 27). Similarly, blocking ACh-mediated currents with 10 nM Cy5-α-BgTx only yielded 2 ± 1% (*n* = 41) of the control current, which was restored when this synthetic peptide was preincubated with 1 μM BSpep to 101 ± 2% (*n* = 44) (Figure 5B). The possibility of an α-BgTx-independent effect of 1 μM BSpep on the ACh-mediated response was also investigated by incubating BSpep with the cells. This resulted in no significant increase from the control current (125 ± 11%, *n* = 14). Overall, these results validate the expected effect of BSpep, occurring through the capture of α-BgTx, preventing its inhibitory action on the muscle-type nAChR. The data also indicate that the inhibition of mAChR by the synthetic Cy5-α-BgTx occurs through the binding of the peptide on the same mAChR site as native α-BgTx. 

### 2.6. Characterisation of the Cellular Distribution of Muscle-Type mAChR with Fluorescent Cy5-α-BgTx

Using fluorescent Cy5-α-BgTx, we aimed at visualising the distribution of mAChR on fixed TE671 cells by confocal microscopy. At a Cy5-α-BgTx concentration of 200 nM, a fluorescent signal was observed at the expected excitation/emission wavelengths (638/700 nm, respectively), although fluorescence was mostly observed intracellularly (Figure 6A). It has previously been reported that TE671 express mAChR as an immature foetal form, known to localise preferentially in intracellular pools, accounting for up to 80% of the total count [47,48,49]. To validate the specificity of the observed fluorescence, Cy5-α-BgTx labelling was also performed after neutralising the toxin with 100 µM BSpep through an overnight incubation at 4 °C. This significantly lowered the recorded cell fluorescence, reaching only 41 ± 6% (*n* = 3 regions of interest; total cell number = 973) of the signal obtained with free Cy5-α-BgTx (*n* = 2 regions of interest; total cell number = 707; *p* < 0.05) (Figure 6B). This result suggests that the major part of the signal observed with the toxin alone is mAChR-specific, whereas the remaining part may also be due to peptide complex internalization or simply an unwanted dissociation of Cy5-α-BgTx from BSpep, occurring as a result of competition with mAChR localized at the plasma membrane. The affinity of BSpep for Cy5-α-BgTx being significantly lower than the one of mAChR dissociation from the peptide is likely a contributing factor. Altogether with its previously discussed electrophysiological characterisation, this contributes to validate the use of Cy5-α-BgTx as a highly potent and selective fluorescent probe for the visualisation of muscle-type mAChR. 

## 3. Conclusions

With 74 amino acids and no less than five disulphide bridges, the chemical synthesis of α-BgTx represents a real technical challenge for two reasons. First, the length of the peptide is considerable, and therefore its chemical synthesis requires a strategy of peptide fragment assembly. Second, the presence of 10 cysteine residues theoretically allows for no less than 45 combinations of disulphide bridge arrangements. Our data indicate that we were able to successfully assemble α-BgTx sequence using three carefully chosen peptide fragments and using two successive hydrazide-based ligations of toxin peptide fragments. Thanks to the addition of an alkyne function at the N-terminus of the toxin, a regioselective addition of an azide-modified fluorochrome was performed. In addition, the linear peptide underwent a proper oxidative folding; a single RP-HPLC peak arose from the process, demonstrating that a single combination of disulphide bridge pairing was largely favoured, without further experimental control. We assume that we naturally reached the proper pattern of disulphide bridging, although we did not check this point specifically. Obviously, it appears that the additional presence of pentynoic acid at the N-terminus did not interfere with the folding process. To ensure that α-BgTx was properly synthesised and that the addition of a fluorescent tag at the N-terminus by click chemistry did not prevent bioactivity, we performed a number of functional tests with Cy5-α-BgTx. We used an automated patch-clamp system along with an optimized protocol to assess the inhibition of ACh-mediated currents on TE671 cells. The results illustrate a slight increase in the IC_50_ value for the synthetic fluorescent α-BgTx compared to the IC_50_ value produced by purified α-BgTx, but this difference can easily be attributed to the difficulties in reliably evaluating the concentration of purified peptide and/or the impact of the fluorescent probe on the activity of the synthetic compound. The difference in the IC_50_ values observed was, however, quite reasonable and indicates complete bioactivity of the synthetic product. Of course, it may be of interest in the future to perform a full comparison between native, non-labelled synthetic and tagged-synthetic α-bungarotoxin to check how the fluorescent tag affects activity directly or indirectly. In a study examining the effect of the fluorescent tag on the pharmacology of protoxin II, it was shown that ATTO488 fluorescent tag was more conservative in the native bioactivity of the peptide than the Cy5 tag, indicating that the nature of the tag matters [50]. Here, the use of Cy5 may yield the same result as protoxin II; however, this can be tested later, as the use of click chemistry allows for an easy change in the tag identity. Remarkably, Cy5-α-BgTx inhibition of ACh-mediated currents could be blocked by preincubation with the BSpep, which mimics the mAChR binding site of the peptide. These results further highlight that the synthetic peptide, similarly to the purified one, blocks mAChR by binding onto a specific and well-defined site on the receptor. The applicability of Cy5-α-BgTx for labelling mAChR was further demonstrated by labelling a receptor on TE671 cells, labelling that could be partially prevented by incubation with the BSpep inhibitory sequence. The synthesis described herein will mainly be of interest for those that need to modify natural amino acids within α-bungarotoxin at the expense of an increased production cost of about five to seven-fold compared to labelled-purified α-bungarotoxin.

## 4. Materials and Methods

### 4.1. Reagents and Materials

All chemical reagents and solvents were purchased from Sigma-Aldrich (Saint-Quentin-Fallavier, France) except the 2-chlorotrityl chloride resin, which was purchased from Iris Biotech GmbH (Marktredwitz, Germany).

### 4.2. Resin Loading for Solid-Phase Peptide Syntheses

Two types of resin loading were performed for the synthesis of α-bungarotoxin. First, we loaded 2-chlorotrityl chloride resin for carboxylated peptides. The first amino acid was loaded manually by treating 1 g of 2-chlorotrityl chloride resin (1.6 mmol/g of Cl) swollen in DCM with a solution of 0.6 mmol amino acid and 1 mL of iPr_2_Net in 5 mL of DCM for 1 h, followed by adding 5 mL of MeOH and agitating for an additional 1 h. The resin was washed twice with DCM and twice with DMF. Next, it was treated three times with 10 mL of a 20% piperidine solution in DMF. The resulting liquid phases were poured together, and the loading was determined by measuring the absorbance of dibenzofulvene-piperidine adducts at 299 nm (ε = 7800 L·mol^−1^·cm^−1^). Second, we loaded 2-chlorotrityl chloride resin for hydrazide peptides. A total of 1 g of 2-chlorotrityl chloride resin (1.6 mmol/g of Cl) was washed twice with DCM and twice with NMP and treated with 10 mL of a 5% hydrazine hydrate solution in NMP for 1 h. The resin was filtered and treated with 10 mL of MeOH/NH_2_NH_2_/NMP 10:5:85 for 30 min, filtered again, and washed twice with NMP, twice with DCM and twice with NMP. The first amino acid was loaded manually by treating the resin with a solution of 0.6 mmol amino acid, 0.6 mmol HCTU and 1.2 mmol of iPr2Net in 5 mL of NMP for 1 h. The resin was filtered and treated with 10 mL of Ac_2_O/iPr_2_Net/NMP 10:5:85 for 15 min, and then filtered and washed twice with DCM and twice with DMF. The resin was treated three times with 10 mL of a 20% piperidine solution in DMF, the resulting liquid phases were poured together, and the loading was determined by measuring the absorbance of dibenzofulvene-piperidine adduct at 299 nm.

### 4.3. Solid-Phase Peptide Syntheses

Solid-phase peptide syntheses (SPPS) of α-bungarotoxin fragments and BSpep (GSGWRYYESSLEPYPDGSGK(biotin)-OH) were run on an automated Symphony synthesizer from Gyros Protein Technologies using Fmoc/tBu chemistry at a 0.1 mmol scale, with HCTU as coupling reagent and 2-chlorotrityl chloride resin as solid support. The elongation was carried out using a 10-fold excess of protected amino acids and HCTU and 20-fold excess of iPr_2_NEt in NMP. The side chain protecting groups used were Asn(Trt), Arg(Pbf), Cys(Trt), Gln(Trt), Glu(O*t*Bu), Lys(Boc), Ser(*t*Bu), Thr(*t*Bu) and Tyr(*t*Bu). Fmoc deprotection was performed using a 20% piperidine solution in DMF. Each coupling lasted 15 min and was repeated three times, followed by capping with a 10% acetic anhydride in NMP. The crude peptides were released from the resin with TFA/iPr3SiH/1,3-dimethoxybenzene/2,2′-(ethylenedioxy)diethanethiol/H_2_O (85:2.5:3.75:3.75:5) for 3 h, and the peptides were precipitated with ice-cold diethyl ether, recovered by centrifugation and washed twice with diethyl ether. HPLC and MS analyses were performed on the crude peptides before purification.

### 4.4. HPLC and Mass Spectrometry Analyses

The peptides were analysed by HPLC and high-resolution ESI-MS mass spectrometry. HPLC analyses were carried out on an Agilent system equipped with a Chromolith^®^ High Resolution RP-18e column (150 Å, 100 × 4.6 mm) at a flow rate of 3 mL/min, with a XSelect Peptide CSH C18 column (130 Å, 2.5 μm, 150 × 4.6 mm) or with an Agilent AdvanceBio Peptide column (2.7 μm, 100 × 2.1 mm or 250 × 2.1 mm) at a flow rate of 1 mL/min and 214 nm. Solvents A and B were 0.1% TFA in H_2_O and 0.1% TFA in MeCN. The acquisition of the LC-ESI-MS data was carried out on a Waters Q-TOF Xevo G2S mass spectrometer with an Acquity UHPLC system and Lockspray source equipped with a Waters Acquity UPLC BEH300 C18 column (1.7 μm, 2.1 × 150 mm). Peptide elution was performed at a flow rate of 0.4 mL/min with a 10–70% gradient of buffer B over 10 min. Solvents A and B were 0.1% formic acid (FA) in H_2_O and 0.1% FA in MeCN. All calculated and found masses were monoisotopic. Purification of the peptides was performed by HPLC on a preparative Agilent 1260 system equipped with a Phenomenex Jupiter column (4 µm Proteo 90 Å, C12, 250 × 4.6mm) at a flow rate of 20 mL/min or a semi-preparative Agilent 1260 system equipped with an Agilent Poroshell EC-C18 column (120 Å, 4 µm, 250 × 9.4 mm) at a flow rate of 4.5 mL/min.

### 4.5. Reduced PA-α-BgTx Synthesis by Native Chemical Ligation

PA-α-BgTx_1-43_ (**6**)—10 mg of PA-α-BgTx_1-28_ (**1**, 1 equiv.) was dissolved in 500 µL of NCL buffer consisting of 200 mM NaH_2_PO_4_ and 6 M guanidinium hydrochloride (Gn.HCl) at pH 3.0 and cooled in an ice/salt bath at −15 °C for 5 min. Then, 56 µL of 0.5 M aqueous NaNO_2_ solution (10 equiv.) was added, and the mixture was agitated for 15 min at −15 °C. A total of 47 mg of MPAA (100 equiv.) was dissolved in 300 µL of NCL buffer at pH 7.0 and added to the reaction mixture, followed by 6.6 mg of α-BgTx_29-43_ (**3**, 1.3 equiv.) dissolved in 500 µL of NCL buffer at pH 3.0. The pH was adjusted to 6.8, and the reaction was agitated overnight under inert atmosphere at room temperature. Then, 80 mg of TCEP (100 equiv.) was dissolved in 8 mL of 6 M Gn.HCl and the pH was adjusted to 5.0; this solution was added to the reaction mixture and mixed for 30 min. The pH was lowered with 500 µL of TFA, and the mixture was washed 3 times with 20 mL of Et_2_O and purified by preparative RP-HPLC using a linear 20–70% buffer B gradient over 40 min; **6** was not pure enough to determine the ligation yield.

Acm-α-BgTx_29-74_ (**8**)—18.2 mg of Acm-α-BgTx_29-43_ (**7**, 1 equiv.) was dissolved in 1 mL of NCL buffer at pH 3.0 and cooled in an ice/salt bath at −15 °C for 5 min. Then, 193 µL of 0.5 M aqueous NaNO_2_ solution (10 equiv.) was added, and the mixture was agitated for 15 min at −15 °C. A total of 162 mg of MPAA (100 equiv.) was dissolved in 500 µL of NCL buffer at pH 7.0 and added to the reaction mixture, followed by 43 mg of α-BgTx_44-74_ (**5**, 1.1 equiv.) dissolved in 1 mL of NCL buffer at pH 3.0. The pH was adjusted to 6.8, and the reaction was agitated overnight under inert atmosphere at room temperature. Then, 277 mg of TCEP (100 equiv.) was dissolved in 10 mL of 6 M Gn.HCl and the pH was adjusted to 5.0; this solution was added to the reaction mixture and mixed for 30 min. The pH was lowered with 500 µL of TFA, and the mixture was washed 3 times with 20 mL of Et_2_O and purified by preparative RP-HPLC using a linear 20–70% buffer B gradient over 40 min. After lyophilization, 33.2 mg of **8** was obtained (yield 60%). RP-HPLC retention time (RT) was 6.5 min (AdvanceBio Peptide C18 10 cm, 5–65% B in 12 min); ESI-HRMS C_207_H_334_N_62_O_68_S_7_ calculated 5000.26; found 5000.29.

α-BgTx_29-74_ (**9**)—27.5 mg of **8** was dissolved in 1 mL of NCL buffer at pH 7.0 and 10 mg of PdCl_2_ (10 equiv.) in 200 µL of NCL buffer at pH 7.0 was added. The mixture was agitated 10 min at 37 °C, and 85 mg of DTT (100 equiv.) in 200 µL of water was added. The reaction mixture was centrifugated, the pellet was washed with 200 µL of NCL buffer, and the liquid phases were poured together, acidified with formic acid, and purified by preparative RP-HPLC using a linear 10–60% buffer B gradient for 40 min. After lyophilization, 14.5 mg of **9** was obtained (yield 53%). RP-HPLC RT 7.2 min (AdvanceBio Peptide C18 10 cm, 5–65% B in 12 min); ESI-HRMS C_204_H_329_N_61_O_67_S_7_ calculated 4929.23; found 4929.16.

Reduced PA-α-BgTx (**10**)—16.8 mg of PA-α-BgTx_1-28_ (**1**, 1.5 equiv.) was dissolved in 600 µL of NCL buffer at pH 3.0 and cooled in an ice/salt bath at −15 °C for 5 min. Then, 95 µL of 0.5 M aqueous NaNO_2_ solution (10 equiv.) was added, and the mixture was agitated for 15 min at −15 °C. A total of 79 mg of MPAA (100 equiv.) was dissolved in 200 µL of NCL buffer at pH 7.0 and added to the reaction mixture, followed by 18.0 mg of α-BgTx_29-74_ (**9**, 1 equiv.) dissolved in 600 µL of NCL buffer at pH 3.0. The pH was adjusted to 6.8, and the reaction was agitated overnight under inert atmosphere at room temperature. Then, 135 mg of TCEP (100 equiv.) was dissolved in 6 mL of 6 M Gn.HCl and the pH was adjusted to 5.0; this solution was added to the reaction mixture and mixed for 30 min. The pH was lowered with 500 µL of TFA, and the mixture was washed 3 times with 20 mL of Et_2_O and purified by preparative RP-HPLC using a linear 20–70% buffer B gradient for 40 min. After lyophilization, 12.3 mg of **10** was obtained (yield 42%). RP-HPLC RT 9.1 min (AdvanceBio Peptide C18 10 cm, 5–65% B in 12 min); ESI-HRMS C_343_H_543_N_97_O_106_S_11_ calculated 8068.70; found 8068.68.

### 4.6. Folding of Reduced PA-α-BgTx

A total of 3.0 mg of reduced PA-α-BgTx (**10**) was dissolved in 3 mL of water and added dropwise to 30 mL of a 0.1 M Tris.HCl buffer at pH 8.2 containing 5 mM GSH and 0.5 mM GSSG while stirring. After 48 h at room temperature, the reaction mixture was acidified with 1 mL of formic acid, filtered and purified by semi-preparative RP-HPLC using a linear 20–40% buffer B gradient over 30 min at 40 °C. After lyophilization, 680 µg of oxidized PA-α-BgTx (**11**, quantified by UV measurement, ε = 9105 L·mol^−1^·cm^−1^) was obtained (yield 23%). RP-HPLC RT 21.1 min (AdvanceBio Peptide C18 25 cm, 5–65% B in 12 min); ESI-HRMS C_343_H_533_N_97_O_106_S_11_ calculated 8058.62; found 8058.60.

### 4.7. Coupling of Cy5 on PA-α-BgTx

A total of 110 µg of oxidized PA-α-BgTx (**11**, 1 equiv.) was dissolved in 12 µL of (NMP/*t*BuOH 9:1)/H_2_O/(0.4 M HEPES, 50 mM aminoguanidine hydrochloride) 1:1:1, and 0.1 mg of Cy5-N_3_ (**12**, 10 equiv.) dissolved in 10 µL of NMP/*t*BuOH 9:1 was added. Then, 2.8 mg of CuBr.(CH_3_)_2_S and 3 mg of THPTA were dissolved in 1 mL of NMP/*t*BuOH; 10 µL of this solution was added to the reaction mixture (10 equiv. of CuBr.(CH_3_)_2_S and 5 equiv. of THPTA) and agitated overnight under inert atmosphere at room temperature. The reaction mixture was acidified with 1 drop of formic acid and purified by semi-preparative RP-HPLC using a linear 20–70% buffer B gradient for 30 min at 40 °C. After lyophilization, 25 µg of Cy5-α-BgTx (**13**, quantified by UV measurement, ε = 250,000 L·mol^−1^·cm^−1^ at 646 nm) was obtained (yield 21%). RP-HPLC RT 16.9 min (XSelect Peptide CSH C18, 5–60% B in 20 min); ESI-HRMS C_378_H_578_N_103_O_107_S_11_ calculated 8623.99; found 8623.98.

### 4.8. Cell Culture

TE671 cells were maintained in Dulbecco’s Modified Eagle’s Medium (DMEM)-high glucose (4.5 g/L) supplemented with 10% foetal calf serum, 1 mM pyruvic acid, 4 mM glutamine, 10 IU/mL penicillin and 10 μg/mL streptomycin (Gibco, Grand Island, NY, USA), and incubated at 37 °C in a 5% CO_2_ atmosphere. For electrophysiological recordings, cells were detached with trypsin, and floating single cells were diluted (~300 k cells/mL) in extracellular medium containing (in mM): 4 KCl, 140 NaCl, 5 Glucose, 10 HEPES, 2 CaCl_2_, 1 MgCl_2_ (pH 7.4, osmolarity 298 mOsm).

### 4.9. Ligand Binding Studies

Binding of BSpep to Cy5-α-BgTx was investigated by microscale thermophoresis (MST) using the Monolith NT 115 (NanoTemper Technologies GmbH, Germany). All dilutions were performed in proprietary MST NT.115 buffer, supplemented with 0.05% Tween 20. Briefly, 5 nM of Cy5-α-bungarotoxin was incubated with a range of BSpep concentrations for 15 min at room temperature, before being loaded in Monolith Premium capillaries (NanoTemper Technologies GmbH, Germany) and mounted in the Monolith NT 115 thermophoresis chamber. MST was monitored using the dedicated MO Control software. Readings were performed using the built-in picoRED illumination source (600 to 650 nm) at 20% illumination power, and MST power was set as medium. MST fluorescence (F1) was averaged from 1.5 to 2.5 min through MST and normalised against initial fluorescence (F0), which was recorded 1 min before the start of MST. The resulting normalised fluorescence values were expressed as the difference from initial fluorescence (ΔFnorm), which was plotted against BSpep concentration. Dose-ΔFnorm relationships were fit to a sigmoidal equation: Y = Bottom + (Top − Bottom)/(1 + 10^((LogIC_50_ − X) × HillSlope)).

### 4.10. Automated Patch-Clamp Recordings

Whole-cell recordings were used to investigate the effects of purified α-bungarotoxin and synthetic Cy5-α-bungarotoxin on ACh-induced currents in TE671 cells, by adapting a protocol previously used by Taiwe et al. [45]. Automated patch-clamp recordings were performed using a SyncroPatch 384PE (Nanion, München, Germany). Chips with single-hole high resistance (~4–5 MΩ) were used for TE671 cell attachment and recordings. Pulse generation and data collection were performed with PatchControl384 v1.6.6_B18 (Nanion) and Biomek v4.1_B31 (Beckman Coulter, Brea, CA, USA). Whole-cell recordings were conducted according to the Nanion procedure. Cells were stored in a cell hotel reservoir at 10 °C, with shaking speed at 60 RPM. After initiating the experiment, cell catching, sealing, whole-cell formation, liquid application, recording, and data acquisition were performed sequentially. The intracellular solution contained (in mM): 10 KCl, 110 KF, 10 NaCl, 10 EGTA and 10 HEPES (pH 7.2, osmolarity 280 mOsm). The extracellular filling solution contained (in mM): 140 NaCl, 4 KCl, 5 glucose and 10 HEPES (pH 7.4, osmolarity 290 mOsm). The extracellular solution contained (in mM): 140 NaCl, 4 KCl, 2 CaCl_2_, 1 MgCl_2_, 5 glucose and 10 HEPES (pH 7.4, osmolarity 298 mOsm). Whole-cell patch experiments were performed at a holding potential of −60 mV at room temperature (20–22 °C). Currents were sampled at 10 kHz. Each molecule, acetylcholine (100 µM, Sigma, Saint-Quentin Fallavier, France), purified α-bungarotoxin and Cy5-α-bungarotoxin and BSpep 1, was prepared in extracellular solution supplemented with 0.3% bovine serum albumin and 0.86 mM Ca^2+^ in 384-well compound plates. Working compound solution was diluted 3 times in recording wells by adding 30 µL compound to 60 µL external solution to reach final concentration. Where required, BSpep was preincubated with native α-bungarotoxin or Cy5-α-bungarotoxin for 2 h at 4 °C. ACh was applied as a 5 s pulse and rapidly washed using stacked addition to avoid excess mAChR desensitisation. In order to obtain single cell-specific controls, an initial pulse of ACh was performed, and the elicited current was recorded, followed by 6 washes with extracellular buffer. Washes were performed as sequential addition and removal of 30 µL extracellular medium. Following these washes, native α-bungarotoxin or Cy5-α-bungarotoxin ± preincubation with BSpep was added and allowed to incubate for 15 min in working solution. A second stacked addition of ACh was then performed, and the current response was recorded. Data were analysed with DataController384 V1.8.0_B24 and GraphPad Prism. Dose-response relationships were fit to a sigmoidal equation: Y = Bottom + (Top − Bottom)/(1 + 10^((LogIC_50_ − X) × HillSlope)).

### 4.11. Cell Imaging by Confocal Microscopy

TE671 were incubated overnight at 37 °C in a 5% CO_2_ atmosphere in DMEM (Gibco, Grand Island, NY, USA) on Ibidi µ-slide 18-well plates (Ibidi GmbH, Germany) in order to obtain a cell confluency of approximately 70% on the day of fixation. Cells were washed with non-supplemented DMEM, followed by fixation with 50 µL 4% paraformaldehyde (PFA) over 5 min. Exposure time to the fixative was closely monitored to minimise cell permeabilization. Fixed cells were then washed with phosphate buffer solution (PBS), prior to labelling with 50 µL staining solution (PBS with 1% Bovine Serum Albumin, 5 µg/mL Hoechst and 200 nM Cy5-α-bungarotoxin). Then, 200 nM Cy5-α-bungarotoxin was preincubated overnight at 4 °C with 100 µM BSpep for competition experiments. Incubation was performed over 30 min at RT. Labelling was followed by 3 PBS washes of 5 min each. Wells were ultimately bathed in PBS + 0.1% PFA. Image acquisition was performed with a NIKON A1 RSi confocal microscope (Nikon, Champigny sur Marne, France), using a ×60 magnification. Hoechst fluorescence was excited at 403 nm and recorded at 450 nm. Excitation/recording wavelengths for Cy5 were 638/700 nm, respectively. Images were analysed, and fluorescence was quantified using ImageJ. 

### 4.12. Statistical Analyses

Data analyses were performed using GraphPad Prism. All data were presented as mean ± s.e.m. Comparison of peak current amplitude was performed by a Kruskal-Wallis test with Dunn’s multiple comparison correction. Comparison of fluorescence intensity was performed by a two-tailed *t*-test, following normality confirmation by a Shapiro-Wilk test. *p* values were considered significant below 0.05. 

## Figures and Tables

**Figure 1 toxins-14-00079-f001:**
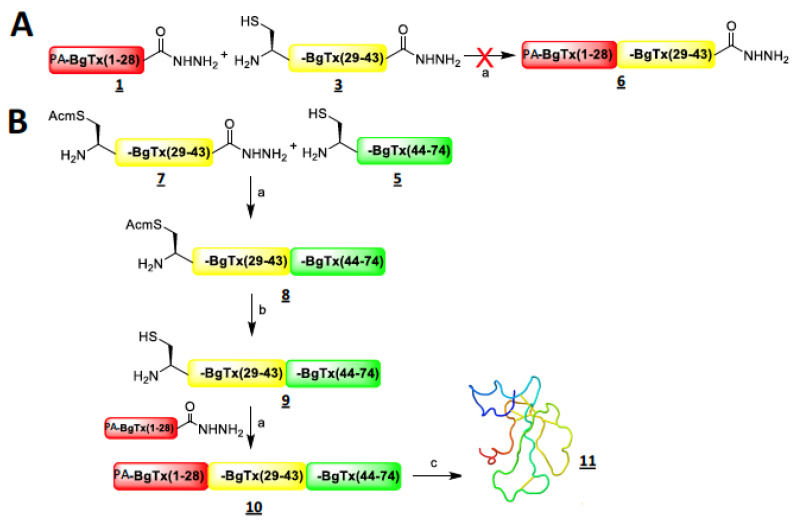
Syntheses strategies of PA-α-BgTx by NCL. (**A**) Synthesis strategy from the N-terminus towards the C-terminus. Only the first step is shown, since it was a dead-end procedure. a = 0.2 M NaH_2_PO_4_, H_2_O, 6 M Gn.HCl, pH 3, −15 °C, 10 eq. NaNO_2_, 20 min, then 100 eq. 4-mercatophenylacetic acid (MPAA), pH 6.5, room temperature (RT), overnight. (**B**) Synthesis strategy from the C-terminus towards the N-terminus. b = Acm deprotection: 0.2 M NaH_2_PO_4_, H_2_O, 6 M Gn.HCl, pH 7.0, 10 eq. PdCl_2_, 10 min, then 100 eq. dithiothreitol (DTT). c = oxidative folding: 0.1 M Tris pH 8.2, 5 mM glutathione (GSH), 0.5 mM glutathione disulphide (GSSG), 48 h.

**Figure 2 toxins-14-00079-f002:**
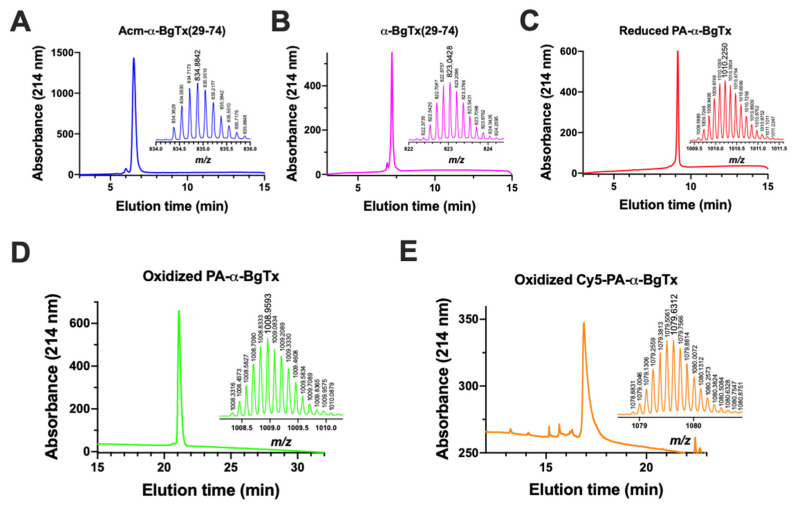
RP-HLPC and MS profiles of synthetic peptides. RP-HPLC A to C were performed on an AdvanceBio Peptide C18 column (10 cm, 214 nm, 5–65% B solvent in 12 min), RP-HPLC D on an AdvanceBio Peptide C18 column (25 cm, 214 nm, 15–65% B solvent in 25 min) and RP-HPLC E on a XSelect Peptide CSH C18 column (15 cm, 214 nm, 5–60% B solvent in 20 min). Retention times were t_R_ = 6.5 min (**A**), 7.2 min (**B**), 9.1 min (**C**), 21.1 min (**D**) and 16.9 min (**E**).

**Figure 3 toxins-14-00079-f003:**
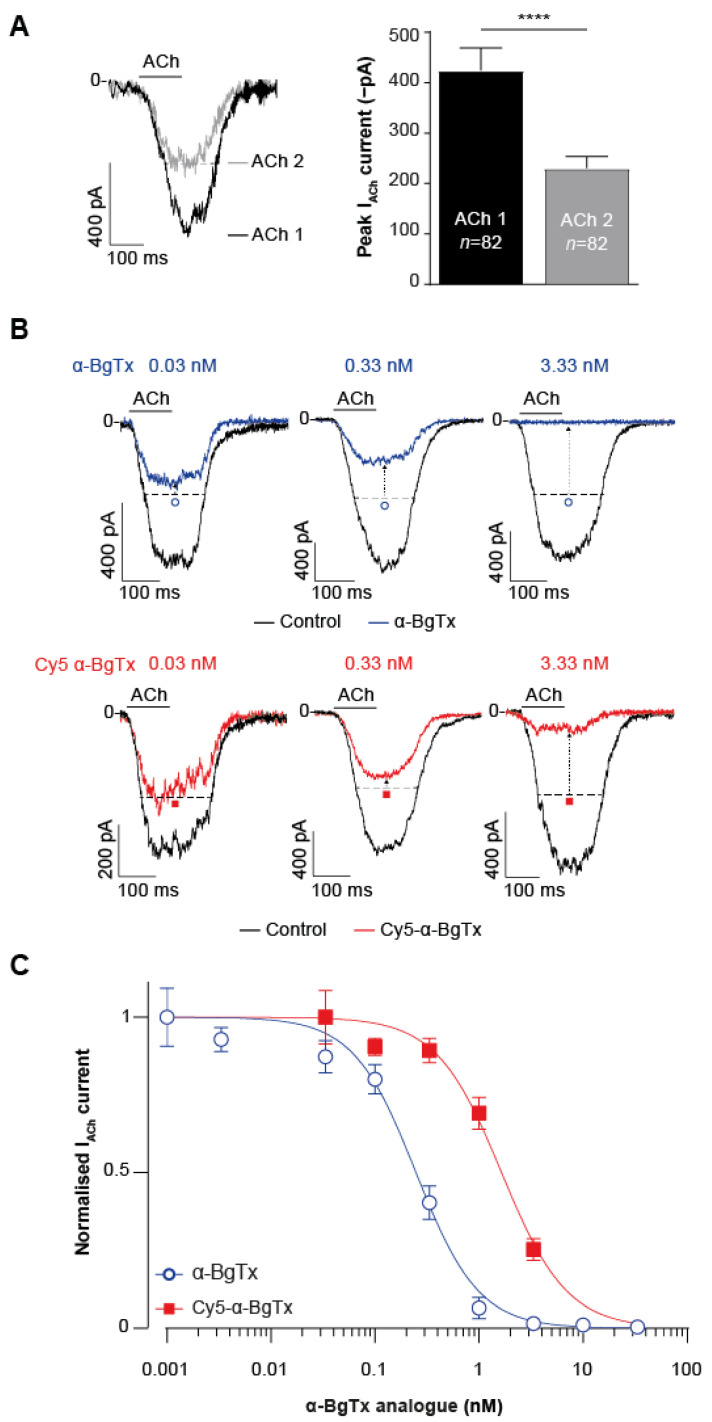
Effect of native α-BgTx and Cy5-α-BgTx on ACh-mediated current in TE671 cells. (**A**) Representative ACh-mediated currents following two consecutive 3.33 μM ACh doses. First ACh pulse illustrated in black, second dose in grey. Mean ± SEM (*n* = 82 cells, **** *p* < 0.0005). The reduction in current amplitude with the second ACh application is due to desensitisation of mAChR. (**B**) Representative current traces in response to a first application of 3.33 ACh (black trace), or a second identical application in presence of increasing concentrations of native α-BgTx (blue) or Cy5-α-BgTx (red). The dotted line represents the expected current after a second ACh application in the absence of any toxin. (**C**) Concentration-response curves for the native α-BgTx and Cy5-α-BgTx-mediated block of ACh response (*n* = 8–41 cells/concentration, total *N* = 339 cells), **** *p* < 0.0005.

**Figure 4 toxins-14-00079-f004:**
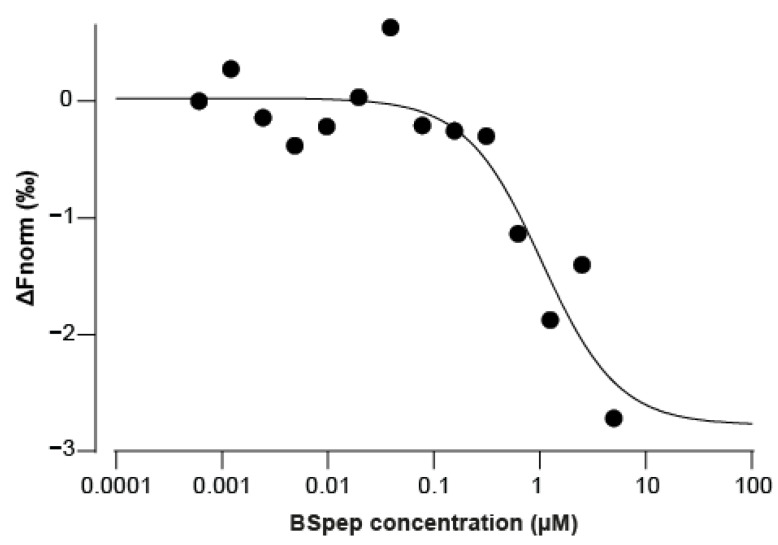
BgTx as measured by MST. A total of 5 nM Cy5-α-BgTx was incubated for 15 min at room temperature with a range of BSpep concentrations. The change in thermophoretic signal yields a K_D_ = 1.04 µM in these experimental conditions.

**Figure 5 toxins-14-00079-f005:**
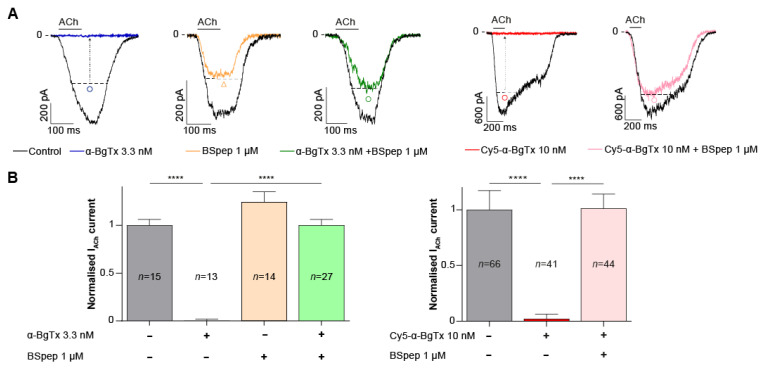
Effect of the BSpep on native α-BgTx and Cy5-α-BgTx inhibition of ACh-mediated current in TE671 cells. (**A**) Representative current traces in response to a first application of 3.33 μM ACh (black) or a second identical application in the presence of α-BgTx (blue), BSpep (yellow), α-BgTx + BSpep (green), Cy5-α-BgTx (red) or Cy5-α-BgTx + BSpep (pink). The dotted line represents the expected current after a second ACh control dose in the absence of toxins. (**B**) Mean normalised current after the second 3.33 μM ACh dose in the presence of α-BgTx (blue), BSpep (yellow), α-BgTx + BSpep (green), Cy5-α-BgTx (red) or Cy5-α-BgTx + BSpep (pink). Mean ± SEM. Kruskal-Wallis test with Dunn’s multiple comparison post-test, **** *p* < 0.0005.

**Figure 6 toxins-14-00079-f006:**
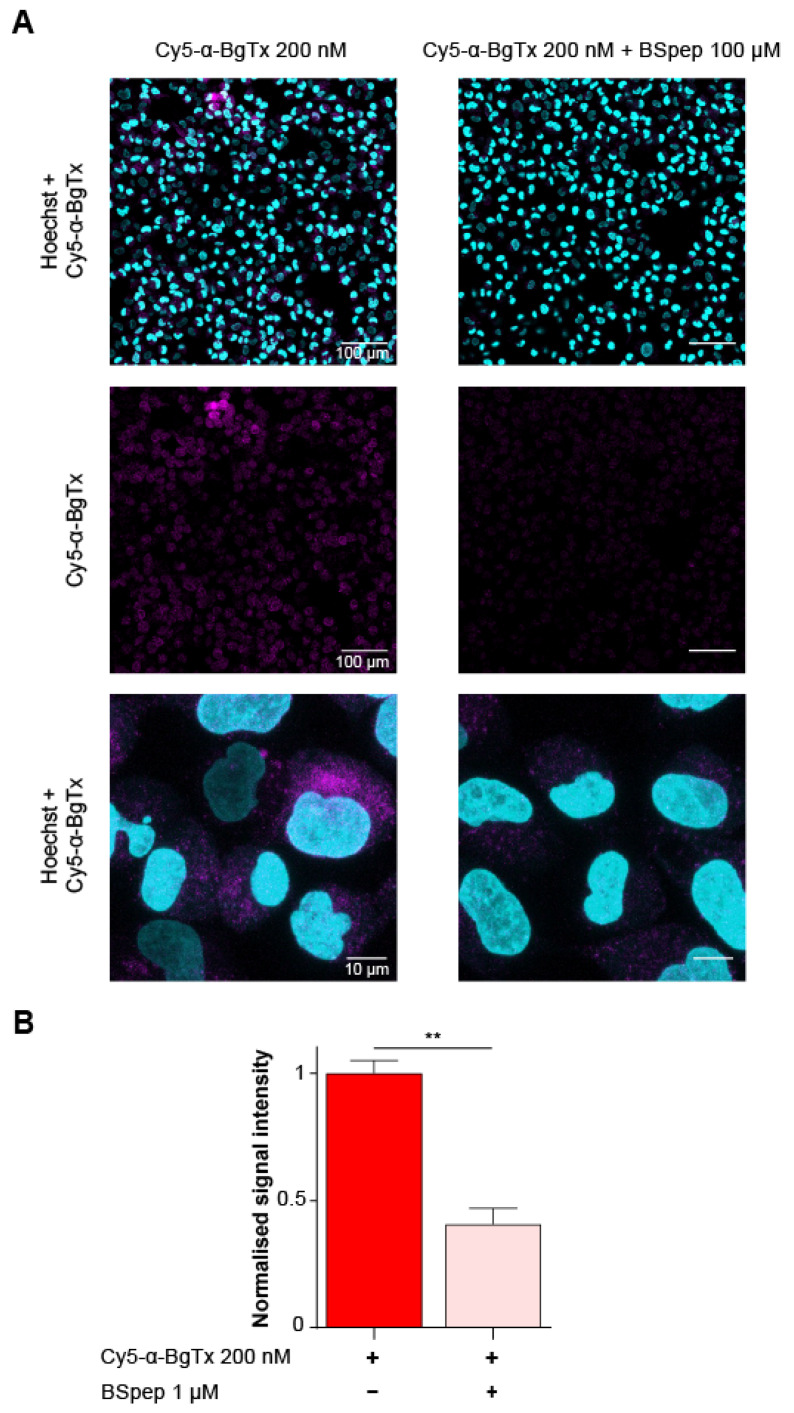
Distribution of nAChR in TE671 cells as labelled with Cy5-α-BgTx. (**A**) Confocal microscopic imaging of TE671 cells labelled with 5 µg/mL Hoechst and either 200 nM Cy5-α-BgTx (left) or 200 nM Cy5-α-BgTx pre-incubated with 100 µM BSpep (right). Top: X60 magnification, Hoechst (cyan) and Cy5-α-BgTx (magenta) displayed; Middle: X60 magnification, only Cy5-α-BgTx displayed; Bottom: X240 magnification, Hoechst and Cy5-α-BgTx displayed. (**B**) Quantification of Cy5-α-BgTx fluorescence after labelling with either 200 nM Cy5-α-BgTx (red) or 200 nM Cy5-α-BgTx pre-incubated with 100 µM BSpep (pink). Mean ± SEM. Two-tailed *t*-test, ** *p* < 0.05.

## Data Availability

Raw data are available on https://github.com/jerome-montnach/Cy5-BgTx.git repository accessed on 2 January 2022.

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
