# Peer review of "Chemical Synthesis of a Functional Fluorescent-Tagged α-Bungarotoxin"

_toxins, 2022, doi:10.3390/toxins14020079_

Round 1

Reviewer 1 Report

Presented manuscript is dedicated to synthesis and characterization of fluorescent labeled alfa-bungarotoxin and its biological evaluation. The manuscript is well-written  and clear for understanding. Only a few remarks were arisen after reading:

- Reference 29 is incorrect; authors should add page numbers and issue number

- The manuscript has detailed introduction concerning alfa-bungarotoxin, its synthesis and applications for AChR studies. To my opinion, it could be interesting to discuss methods of labeling of natural alfa-bungarotoxin with fluorescent markers.

- Various fluorescent labeled natural alfa-bungarotoxins have been described in literature and are presented on market. Is there any published data about effects of fluorescent fragment on activity of bungarotoxin? Did the authors synthesized alfa-bungarotoxin without labels? What about its activity in compearence with labeled on and natural

Author Response

Presented manuscript is dedicated to synthesis and characterization of fluorescent labeled alfa-bungarotoxin and its biological evaluation. The manuscript is well-written and clear for understanding. Only a few remarks were arisen after reading:

- Reference 29 is incorrect; authors should add page numbers and issue number

Response: this reference was indeed incorrect. It has been updated and the correct version introduced in the reference list.

- The manuscript has detailed introduction concerning alfa-bungarotoxin, its synthesis and applications for AChR studies. To my opinion, it could be interesting to discuss methods of labeling of natural alfa-bungarotoxin with fluorescent markers.

Response: We feel that a full discussion would fall a bit outside the scope of this publication. However, we added the following sentence in the introduction “Most of this labeling involves N-hydroxysuccinimide (NHS) ester-mediated derivitization of an amine-reactive group of a-bungarotoxin”. The technologies used are quite classical and stoichiometry of labeling is simply controlled by ratios of concentrations of the interactants. As a result, labelling occurs randomly anywhere where a fee amine is present. In addition, publications report very little details on the methods of labelling making this technical aspect quite obscure in the literature. Finally, most of the labeling techniques are now used by companies that are selling fluorescent bungarotoxin without providing details on how this labelling has been performed.

- Various fluorescent labeled natural alfa-bungarotoxins have been described in literature and are presented on market. Is there any published data about effects of fluorescent fragment on activity of bungarotoxin? Did the authors synthesized alfa-bungarotoxin without labels? What about its activity in comparison with labeled on and natural.

Response: No, we did not find evidence for the importance of the tag on peptide activity but we presume that it may influence. This is a good question. We showed that this is the case for protoxin II for instance. We did not ourselves examine the activities of these commercial bungarotoxins. It could be of interest indeed to perform such a quantitative evaluation and we found no evident study on this topic. Unfortunately, we did not try alpha-bungarotoxin without label and we cannot examine it now because we used all the synthesis for the tagging with Cy5. But it is a good question and as such we discussed this issue in the discussion section.

Reviewer 2 Report

Chemical synthesis of a functional fluorescent-tagged a-bungarotoxin

The authors present a strategy to synthesize α-bungarotoxin, a complex peptide with 74 residues and 5 disulfide bridges. They successfully synthesized the peptide and through various tests verified the activity of the synthetic peptide compared to the original purified from natural sources, obtaining very good results.

I consider that the subject presented is interesting and well developed. Synthesis of the complex peptide is achieved, as is the verification of its functionality.

I have only some minor concerns:

  1. How can the authors ensure that the formation of the disulfide bridges is correct as well as the folding of the peptide? Perhaps the use of circular dichroism could help answer this question.
  2. Although the authors propose chemical synthesis as an alternative to the production of recombinant peptide, due to the possibility of making special modifications, it would be interesting to know the cost of production since this is a factor to consider.
  3. Abbreviations must be defined the first time they appear in the text, including the names of the reagents used.
  4. MS profiles in Fig 2 are not clear, and the values of the peaks are not readable.

Author Response

Chemical synthesis of a functional fluorescent-tagged a-bungarotoxin

The authors present a strategy to synthesize α-bungarotoxin, a complex peptide with 74 residues and 5 disulfide bridges. They successfully synthesized the peptide and through various tests verified the activity of the synthetic peptide compared to the original purified from natural sources, obtaining very good results.

I consider that the subject presented is interesting and well developed. Synthesis of the complex peptide is achieved, as is the verification of its functionality.

I have only some minor concerns:

1. How can the authors ensure that the formation of the disulfide bridges is correct as well as the folding of the peptide? Perhaps the use of circular dichroism could help answer this question.

Response: Circular dichroism would unfortunately not provide the response nor does NMR data. For that purpose, it would require partial enzymatic digestion followed by MS analyses. Misfolding would however not provide such a high activity. We now added a comment in the discussion to state that we did not check for the disulfide bridge assembly pattern.

2. Although the authors propose chemical synthesis as an alternative to the production of recombinant peptide, due to the possibility of making special modifications, it would be interesting to know the cost of production since this is a factor to consider.

Response: Yes of course the cost is higher than simply labeling purified peptide. I would say about 5000 euros for 1 mg, which is an order of 5-7-fold more expensive than the tagged purified toxin. We introduced an estimate of this cost in the manuscript. This is now stated in the discussion of the manuscript.

3. Abbreviations must be defined the first time they appear in the text, including the names of the reagents used.

Response: This is now done.

4. MS profiles in Fig 2 are not clear, and the values of the peaks are not readable.

Response: Figure 2 has been improved by enhancing the font size mainly for the peak of the MS profile.